# Online Detection of Watercore Apples by Vis/NIR Full-Transmittance Spectroscopy Coupled with ANOVA Method

**DOI:** 10.3390/foods10122983

**Published:** 2021-12-03

**Authors:** Yifei Zhang, Xuhai Yang, Zhonglei Cai, Shuxiang Fan, Haiyun Zhang, Qian Zhang, Jiangbo Li

**Affiliations:** 1College of Mechanical and Electrical Engineering, Shihezi University, Shihezi 832003, China; zhangyifei@stu.shzu.edu.cn (Y.Z.); czlczl1024@163.com (Z.C.); gzhshzu@163.com (H.Z.); zqq80@163.com (Q.Z.); 2Intelligent Equipment Research Center, Beijing Academy of Agriculture and Forestry Sciences, Beijing 100097, China; fansx@nercita.org.cn

**Keywords:** watercore apple, online detection, ANOVA analysis, band ratio, threshold discrimination

## Abstract

Watercore is an internal physiological disorder affecting the quality and price of apples. Rapid and non-destructive detection of watercore is of great significance to improve the commercial value of apples. In this study, the visible and near infrared (Vis/NIR) full-transmittance spectroscopy combined with analysis of variance (ANOVA) method was used for online detection of watercore apples. At the speed of 0.5 m/s, the effects of three different orientations (O1, O2, and O3) on the discrimination results of watercore apples were evaluated, respectively. It was found that O3 orientation was the most suitable for detecting watercore apples. One-way ANOVA was used to select the characteristic wavelengths. The least squares-support vector machine (LS-SVM) model with two characteristic wavelengths obtained good performance with the success rates of 96.87% and 100% for watercore and healthy apples, respectively. In addition, full-spectrum data was also utilized to determine the optimal two-band ratio for the discrimination of watercore apples by ANOVA method. Study showed that the threshold discrimination model established based on O3 orientation had the same detection accuracy as the optimal LS-SVM model for samples in the prediction set. Overall, full-transmittance spectroscopy combined with the ANOVA method was feasible to online detect watercore apples, and the threshold discrimination model based on two-band ratio showed great potential for detection of watercore apples.

## 1. Introduction

Watercore is an internal tissue disorder of apples, in which the intercellular air spaces throughout the fruit become filled with fluid, resulting in characteristic translucent tissues [1]. Generally, watercore occurs around the vascular bundles [2] and appears as a water-soaked area [3]. During storage, slight watercore may disappear, while apples with severe watercore may cause browning and alcohol flavors [2,4,5,6], and lost economic value. Some researchers found that the flesh tissues with watercore had higher sorbitol and sucrose concentration and lower glucose concentration [7,8,9]. So, the sweetness and soluble solids content (SSC) of watercore apples are higher. Watercore apples are prevalent in some Asian countries such as China and Japan [5,10]. In Japan, watercore is considered as an indicator of fully mature apples, and such apples are also called “honeyed” apples [2]. However, it is very difficult to identify apples with or without watercore by the naked eyes because the watercore apples have no obvious external symptoms [11]. Therefore, detection of apple watercore can provide not only storage advice to businesses, but also high-quality fruit to consumers.

Watercore can be visually inspected according to the cross-section region after cutting the apple along the equator, however, this way is destructive and costly. Over the years, several non-destructive detection techniques for apple quality detection, such as thermal imaging technology [1], magnetic resonance imaging (MRI) [12], X-ray computed tomography (X-ray CT) [5], and near-infrared hyperspectral imaging [13,14], have been studied. The electrical properties of fruit have also been studied to detect the watercore [15]. Although the above methods can achieve non-destructive detection, these technologies cannot meet the industrial detection demand for mass samples due to being time-consuming and costly [16,17]. Relatively speaking, the visible and near-infrared (Vis/NIR) spectroscopy is a non-destructive, rapid, real-time, and low-cost technology [18,19] for internal quality detection of agricultural products [20]. It has been used for evaluation of fruit quality attributes such as bruise [21,22], defects [23,24], firmness [25,26], SSC [27,28,29], brown [27,30], and moldy core [17,31]. The transmission and diffuse reflection modes are two common ways to collect spectra of tested samples. Compared with the diffuse reflection model, the transmission model has better performance for detecting the internal quality of fruit [32,33]. In a new study, Guo et al. [11] built a detection platform based on near infrared transmission spectroscopy to quantitatively evaluate the watercore degree of apples. Partial least square (PLS) based on 37 effective variables selected by competitive adaptive reweighted sampling (CARS) obtained the optimal performance with coefficient correlation of 0.9562 and root mean square error of 1.34% for the prediction set. This research illustrated that near infrared transmittance spectroscopy was able to distinguish watercore apples. Chang et al. [16] used the online Vis/NIR spectroscopy system to detect watercore apples with a success rate of over 95%. The above studies did not consider the influence of detection orientation on the discrimination results of watercore apples, which cannot meet the needs of industrial applications.

The selection of the characteristic wavelength can eliminate redundancy, noise, and collinearity information that exists in raw spectral data, and improve the efficiency and performance of the detection model. The common variable selection methods include Monte Carlo-uninformative variable elimination (MC-UVE) [34], successive projection algorithm (SPA) [35], competitive adaptive reweighted sampling [36], random frog (RF) [37], and Genetic algorithm (GA) [38], and so on. In addition, some combination methods of variable selection were also used to optimize models, such as MC-UVE-SPA [39], MC-UVE-CARS [40], and CARS-SPA [41]. These methods can obtain good results, but the selected variable number is usually too many, which is not suitable for the development of online detection equipment. Analysis of variance (ANOVA) is one of the most frequently-used and robust statistical methods for analyzing the differences between groups. Tian et al. [42] used the effective wavelengths obtained by one-way ANOVA method to detect the early decay of citrus with a success rate of over 98%. Band ratio calculation is not only a spectral analysis method, but also an effective image processing method. It can reduce spectral difference between samples of the same category and increase contrast between different samples [43,44]. Band ratio method was widely applied for inspecting fruit quality [45,46,47]. Lee et al. [48] identified the watermelon seeds infected by bacteria *Acidovoraxcitrulli* based on the optimal Raman hyperspectral band ratio determined by ANOVA. The success rates of healthy seeds and infected seeds were 75% and 86%, respectively. These studies indicated that one-way ANOVA and band ratio could be used to detect fruit quality with fewer variables. However, in previous studies, these two methods were almost used in hyperspectral imaging. In this study, the ANOVA method combined with Vis/NIR full-transmittance spectrum was used to realize the online detection of watercore apples. On the one hand, ANOVA was used to select the characteristic wavelengths to optimize the models; on the other hand, a simple watercore apple detection model was established based on the optimal two band ratio determined by ANOVA and the threshold discrimination method.

In general, the overall goal of this study was to observe the potential of using Vis/NIR full-transmittance spectroscopy coupled with ANOVA method for online detection of watercore apples. Specific objectives were: (1) to explore the influence of three different detection orientations on the detection accuracy of watercore apple; (2) to identify the significant Vis/NIR transmittance spectrum peaks based on F-values of an ANOVA; (3) to develop a threshold discrimination model based on the optimal two-band ratio; (4) to compare the performance of different models.

## 2. Materials and Methods

### 2.1. Apples

Apple samples without any bruises and defects on the surface were purchased from the local fruit and vegetable wholesale market in Beijing of China in March 2021. A total of 265 “Fuji” samples with equatorial diameter of about 80–100 mm were used in this study. All samples were wiped and labeled. To reduce the effect of temperature on the detection accuracy, all samples were placed in a laboratory (20 °C and relative humidity of 60%) for more than 24 h. After obtaining the spectral data of samples, each apple was cut in half along the cross section to check whether the tested sample is a watercore apple. For more information, please see “Inspection Instructions for Apple” of U.S. Department of Agriculture (PDF file: http://www.ams.usda.gov/grades-standards/apple-grades-standards, accessed on 3 November 2021). The cross-sectional images of the healthy and watercore samples are shown in Figure 1A,B respectively.

### 2.2. Transmittance Spectrum Acquisition

The Vis/NIR full-transmittance measurement system shown in Figure 2 was developed in the intelligent detection laboratory of Intelligent Equipment Research Center, Beijing Academy of Agriculture and Forestry Sciences (Beijing, China). This system was used for obtaining spectral data of all samples. The system consists of the following several units: one high sensitivity spectrograph covering the spectra range of 568–1112 nm with a spectra interval of 0.5 nm, one transmission platform with adjustable speed, one position sensor, the illumination unit including one reflector halogen lamp (FUJI, JCR, 150W, 15V, Tokyo, Japan) and one condensing lens, one dark box for preventing external stray light, and one computer for controlling the system. The illumination unit and the spectrograph were installed on two sides of the conveyor belt, respectively. To study the influence of spectral measurement orientation on the accuracy of online detection of watercore apples, spectral information of all apple samples was collected in three differentiations (O1: Stem-calyx axis is perpendicular to conveyor belt with stem upward; irradiated from equator position by halogen lamp and detected from another side by the spectrometer. O2: Stem-calyx axis horizontal with stem towards halogen lamp; irradiated from stem by halogen lamp and detected from the calyx by the spectrometer. O3: Stem-calyx axis horizontal and parallel to the moving direction of the conveyor belt; irradiated from equator position by halogen lamp and detected from another side by the spectrometer). The three detection directions were presented in Figure 3.

In this study, the speed of the conveyor belt was set to 0.5 m/s (about 5 apples per second). For data acquisition, the apples were placed on the tray with O1, O2, and O3 orientations. When the tested sample passes through the position sensor, the spectrometer would receive a trigger signal from the computer, and then the spectrometer started to obtain and record the multipoint transmittance spectra of sample continuously with a short integration time of 0.005 s until the sample passes through the spectrometer. For each apple, about 35 full-transmittance spectra could be collected.

### 2.3. Preprocessing of Spectral Data

About 30–45 spectral curves can be obtained for each sample due to different sample sizes. The spectral information of the front and rear positions of each sample may be saturated. Therefore, the first three spectra and the last three spectra for each sample are eliminated, and then the average spectrum is obtained as the final spectral information of each sample. In this study, the spectral range is set to 680–1000 nm (1185 bands) considering the low signal-to-noise ratio at the beginning and end of spectra.

### 2.4. ANOVA of Spectral Data

An ANOVA is one of the most frequently and robust used statistical methods for analyzing the differences between groups. The F-values of a one-way ANOVA (note that ANOVA was performed by MATLAB 2018b (The Math Works, Natick, MA, USA) and the average spectrum of each sample was used as the input for ANOVA analysis) were used to find the wavelengths representing statistically significant differences for two groups. The higher the F-value, the more statistically significant the mean separation between groups [49]. This method was used to determine the optimal wavebands combination for distinguishing between healthy and watercore apples. In this study, the wavelengths corresponding to the local maximum of F-value were selected as the effective wavelengths to optimize the detection models.

### 2.5. Discrimination Algorithm of Watercore Apples

Least squares-support vector machine (LS-SVM) was employed to classify watercore from healthy apples. LS-SVM is a reformulation of SVM, which can deal with linear and nonlinear multivariable analysis [50]. Compared with SVM, LS-SVM could reduce the complexity of calculation and shorten the data processing time, so as to improve the analysis ability of high-dimensional data [47]. The radial basis function (RBF) was used as the kernel function for the development of the LS-SVM model, because the RBF kernel function has good performance in dealing with the nonlinear relationship between spectra and target category [25,47]. All the calculations of LS-SVM were performed in MATLAB 2018b (The Math Works, Natick, MA, USA) with the free LS-SVM v1.8 toolbox (Suykens, Leuven, Belgium). The starting points of regularization parameter γ and RBF kernel parameter σ^2^ were determined by the coupled simulated annealing (CSA) method. The optimal parameter combination of γ and σ^2^ was determined by grid research method combined with 10-fold cross validation.

### 2.6. ANOVA of Spectral Band Ratio

Band ratio calculation is not only a spectral analysis method, but also an effective image processing method. It can reduce the spectral difference between samples of the same category and increase the contrast between different samples [43,44]. ANOVA of the band ratio has been widely used to detect fruit quality using spectral image [45,46,47,51,52]. However, few studies have used this method to analyze spectral data. This study attempts to use the transmission spectrum band ratio and threshold method to detect watercore apples. The two-band ratio was performed as the following equation:(1)$$R_{i/k}=\frac{T_{i}}{T_{k}},$$
where $R_{i/k}$ represents a quotient of transmittance spectra, $T_{i}$ and $T_{k}$ are the transmission intensities at i nm and k nm band.

The spectrum of each sample includes 1185 wavelengths, so there are 1185 × 1185 possible two-band ratios. One-way ANOVA method was employed to select the two-band combination for classifying the healthy and watercore apples by comparing the F-values of all band ratios between watercore and healthy spectra in the calibration set. According to the highest F-value of all two-band ratios, the optimal band ratio was selected because a larger F-value indicated statistically greater discrimination between the watercore and healthy samples. Following the two-band ratio selection, the optimal threshold was determined at the point of the highest discrimination accuracies for the total samples in the calibration set.

## 3. Results and Discussion

### 3.1. Statistics of Samples

In this study, all apples were divided into two classes including healthy and watercore. Finally, 138 and 127 apples were identified as healthy and watercore samples, respectively. For each class, all apples were divided into calibration and prediction sets based on the SPXY algorithm [53] according to a proportion of 3:1. Table 1 shows class assignment and partition of sample sets for the establishment of classification models. The calibration set consists of 199 samples, including 105 healthy apples and 95 watercore apples. The prediction set consists of 66 samples, including 34 healthy apples and 32 watercore apples.

### 3.2. Raw Spectral Features

Near-infrared radiation has the absorption, reflection, and transmission interaction with the tested sample, and the change of energy can reflect information of the detected substance [54]. Figure 4A–C shows the original average spectra (solid line) and deviation distribution (shadow region) of O1, O2, and O3 orientations, and the red part represents the healthy samples and the blue part represents the watercore samples, respectively. It is obvious that the spectral intensity and trend of healthy apples are almost the same in three different orientations. The transmittance spectral intensity of healthy samples increased from 680 nm to about 920 nm most likely because light in the near-infrared region has better penetration. The spectral trends were similar with those spectra from studies of Tian et al. [55] and Xia et al. [56]. The spectral intensity peak at about 920 nm might be relative to the third overtone of C-H [56]. It should be noted that the spectral trend of healthy samples has a slight peak at 720 nm in the O2 orientation, which is the same as that of the watercore samples. It may be related with the tissue information of the stem and calyx ends. It can be found from Figure 4A,C that the spectral trends of watercore apples in the O1 and O3 orientations are similar, with peaks at 720 nm and 810 nm. Different from O1 and O3 orientation, the spectral intensity of watercore samples was significantly higher than that of healthy samples at 680–850 nm in the O2 orientation. This may be due to the watercore tissue being distributed around the fruit core, which is conducive to the collection of watercore information in the O2 orientation, and that the healthy tissue absorbed more incident light than the watercore tissue. In terms of three different orientations, the spectral intensity and trend of healthy samples and watercore samples show differences, which provided the possibility for the detection of watercore apple by near-infrared transmission spectroscopy. However, spectral overlap between the two kinds of apples has always existed, thus, the latent spectra information should be further analyzed for identification of watercore apples.

### 3.3. Comparison of Full-Spectrum Models Established Based on Different Orientations

Table 2 summarized the classification results of LS-SVM models established based on the spectral data of different acquisition orientations (O1, O2, and O3). It can be clearly seen that the LS-SVM model could effectively distinguish watercore apples from healthy apples. However, it should be noted that the detection accuracy has an obvious difference in terms of different detection orientations. Comparing three orientations, it can be found that O3 direction was most suitable for online detection of watercore apples. In the O3 orientation, the detection accuracy of LS-SVM model for the calibration and prediction sets was 100% and 98.48%, respectively. The results showed that healthy apples and watercore apples can be almost accurately distinguished in the O3 orientation. The discrimination performance of the model based on the spectral data of O1 orientation was slightly inferior to the O3 orientation. The detection success rate of watercore samples was 90.62%. Compared with O1 and O3 orientations, the detection model based on O2 orientation obtained the worst discrimination results, and the recognition success rate of the prediction set was only 90.91%.

### 3.4. Characteristic Wavelength Selection

In order to minimize the number of spectral bands and improve the detection efficiency, the ANOVA method was used to determine the characteristic wavelengths. ANOVA is a collection of statistical models used to analyze the difference between group means such as the variation among and between the different two groups [57]. To find characteristic wavelengths for differentiating between watercore apples and healthy apples, F-values of one-way ANOVA were calculated for spectra collected from the two types of samples. A larger F-value indicates that the statistical mean separation between two groups is more significant. Referring to the research results of Lee [48], this study selects the local maximum F-value to determine the characteristic wavelength for detecting water core apple. Figure 5A–C shows the F-values of all single wavebands for O1, O2, and O3 detection orientations, respectively. Based on the local maximum F-values, as marked in Figure 5, the characteristic wavelengths were selected for each detection orientation as follows: 705.66 nm and 929.76 nm wavebands for O1 orientation, 712.37 nm and 806.23 nm wavebands for O2 orientation, and 704.59 nm and 924.62 nm wavebands for O3 orientation. The bands selection for O1 and O3 orientations was similar, while that for O2 orientation was quite different from those of O1 and O3 orientations. This is related to the intensity distribution of the original spectral data. These selected bands were statistically significant [44,51]. In general, only two characteristic wavelengths were selected for each orientation by the ANOVA method, which greatly reduces the number of variables compared with full spectral data (1185 bands).

### 3.5. Classification Results Based on Characteristic Wavelengths

The characteristic wavelengths selected by ANOVA method were used as the input to establish LS-SVM discrimination model. The discrimination results of watercore apples for each orientation were shown in Table 3. For O1 orientation, the recognition success rates of healthy apples and watercore apples in the prediction set were 94.12% and 93.75%, respectively. The detection results of watercore apples were better than the full-spectrum LS-SVM model, but the discrimination performance of the total samples was inferior to the full-spectrum LS-SVM model. The LS-SVM model built based on the characteristic wavelength selected from the spectral data of O3 orientation has the optimal performance with a detection accuracy of 98.48%, which was the same as the detection result of the full spectrum LS-SVM model. In other words, the O3 direction was the most suitable for detecting watercore apples. In terms of O2 orientation, the prediction accuracy of the total samples in the prediction set was the worst, with an accuracy of only 89.39%. However, the detection performance of healthy apples was superior to O1 orientation. It is worth to note that the O2 orientation obtained the worst recognition result for watercore apple, and the detection success rate was only 81.25%.

For three different detection orientations, the performance of the LS-SVM model based on the characteristic wavelength was slightly inferior to that of the full-spectrum LS-SVM model. In particular, for O3 orientation, the characteristic bands LS-SVM and the full-spectrum LS-SVM models have the same performance to distinguish watercore apples. However, the used variable number in the model was reduced from 1185 bands to 2 bands, which greatly reduces the model complexity. The result shows that the one-way ANOVA method was an effective wavelength selection tool. The watercore apples discrimination results obtained from full-spectrum LS-SVM model and characteristic wavelength LS-SVM model indicated that O3 orientation was the optimal direction.

### 3.6. Classification Results of Spectral Band Ratio

For the healthy and watercore apples in the calibration set, F-values of ANOVA of all two-band ratios were calculated in the range 680–1000 nm and plotted in the contour figure. The largest F-value was selected, and the corresponding two-band ratio was taken as the optimal two-band ratio for discriminating between healthy and watercore samples. Figure 6A–C shows the resultant contour plot of F-values corresponding to O1, O2, and O3 orientations, respectively. The plots show multiple clusters of two-band ratio regions with relatively high F-values (red color). For O1 orientation, the ratios for the wavelengths at 763.90 nm and 718.28 nm produced the largest F-value, indicating that the ratio values of the two groups of samples at these two wavelengths demonstrated the most visible differences. Compared with the peaks in Figure 4A, the 718.28 nm wavelength was near the slight peak of the watercore samples, but the 763.90 nm wavelength did not show any remarkable features. In the O2 orientation, the ratio of the two bands at 756.32 nm and 733.63 nm provided the best result for separating healthy and watercore samples. However, the wavelengths at 756.32 nm and 733.63 nm did not show any remarkable features compared with the main peaks in Figure 4B. The wavelengths corresponding to the optimal two-band ratio in O3 and O1 orientations were very near.

The optimal threshold selection method based on two-band ratio was the same for three different orientations. Therefore, taking the optimal threshold selection of O1 orientation as an example, it is described in detail. According to the distribution of 763.90 nm/718.28 nm two-band ratio for healthy and watercore samples of calibration set (Figure 7A), the distribution of the two groups overlapped with each other when the 763.90 nm/718.28 nm values were between 1.3256 and 1.3743, which resulted in false identification between watercore and healthy apples. Therefore, a proper threshold value was required for discrimination. A two-band ratio below the threshold value indicated that the apples were obtained from watercore samples, whereas a two-band ratio above the threshold value indicated that the apples were obtained from healthy samples. The two-band ratio of watercore and healthy samples were distributed between 0.9124 and 1.3743 and 1.3256 and 1.7007, respectively. In order to clearly explain the influence of threshold value on the detection accuracy for total samples, the two-band ratio values increasing from 0.9124 to 1.7007 with a step of 0.0001 were shown in Figure 7B. The highest point of accuracy lines (1.3743) was determined to be the optimal threshold value. Finally, the classification success rate for the prediction set was obtained by using the selected two-band ratios and threshold. The two-band ratio distribution and classification accuracy curves of watercore samples and healthy samples of the calibration set in the O2 and O3 orientations were shown in Figure 8A,B and Figure 9A,B, respectively.

Table 4 summarized the classification performance of the threshold discrimination model based on the optimal two-band ratio. The threshold discrimination model based on the two-band ratio of O1 orientation obtained a detection success rate of 95.45% for the total samples in the prediction set. The highest overall accuracy of 98.48% for the prediction set was achieved by a threshold discrimination model based on the two-band ratio in the O3 orientation. Comparing different detection orientations, it can be found that O3 orientation is the most suitable for online detection of watercore apples, and O2 orientation obtained the worst detection results. In three different orientations, the threshold discrimination model and LS-SVM model showed almost the same performance. However, the threshold discrimination model only needs two bands, which is of great significance for the development of online detection equipment of watercore apples.

As a comparison, previous studies did not consider the influence of orientation on detection accuracy [16]. In this study, the detection orientation was considered and obtained the higher classification accuracy (98.48%) between healthy and watercore apples. The detection speed (about five apples per second) used in this study was higher than that of Chang’s study (three apples per second). In particular, the internal quality detection of fruit based on the method of ANOVA and two-band ratio combined with Vis/NIR spectroscopy has not been reported. Moreover, the classification accuracy was better than Herremans’s study [5], which used X-ray CT and MRI techniques. Hyperspectral transmittance imaging systems [46,58] were also used to detect the internal defect of fruit. However, the higher implementation costs and lower detection efficiency limited the practical application of hyperspectral technology.

## 4. Conclusions

The Vis/NIR full-transmittance spectroscopy was successfully utilized for online classification of watercore apples. Two characteristic wavelengths were selected by one-way ANOVA and the two-wavelength LS-SVM model was established for classification. Compared with the full-spectrum LS-SVM model, the two-wavelength LS-SVM model reduces the modeling variables and improves the detection efficiency. Results show that the one-way ANOVA method was a powerful tool for selection of characteristic wavelengths. In addition, the two-band ratios of full-spectrum data were also calculated by the ANOVA method to determine the optimal two-band ratio. The threshold discrimination model was built and the accuracy of online detection of watercore apples was higher than 89.89%. It indicated that the proposed two-band ratio threshold discrimination model can obtain good performance for online detection of watercore apples, which provides a new method for the internal quality detection of apples. Comparing the three different detection orientations, it was found that the sample orientation had a significant impact on the classification accuracy of watercore apples. The distribution of watercore tissue may lead to spectral difference in different orientations, which affects the detection results. The O3 orientation was the most suitable for watercore apples detection and the classification accuracy of both LS-SVM model and threshold discrimination model was 98.48%. Vis/NIR full-transmittance spectroscopy combined with ANOVA was a non-destructive and effective method for watercore apples detection. Two characteristic wavelengths determined by the ANOVA method can realize the discrimination of watercore apples, which is significant for designing an effective detection system for practical production in the industry.

## Figures and Tables

**Figure 1 foods-10-02983-f001:**
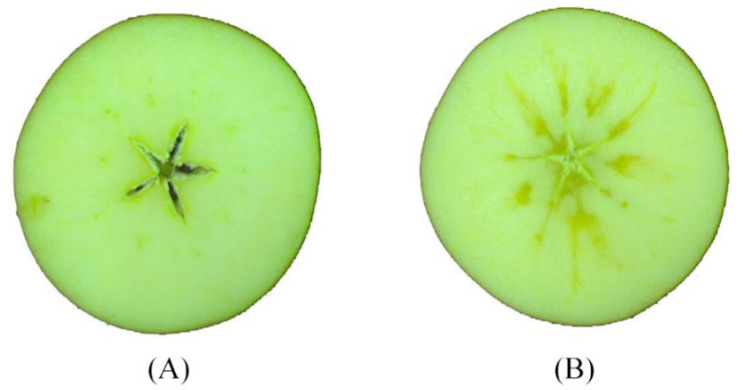
Cross-sectional images of samples: (**A**) healthy apple, (**B**) watercore apple.

**Figure 2 foods-10-02983-f002:**
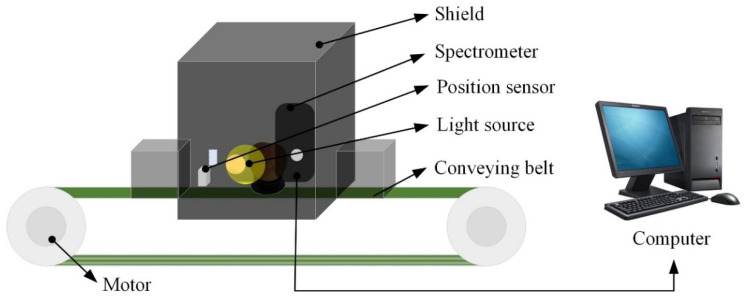
Schematic of full-transmittance spectra measurement system.

**Figure 3 foods-10-02983-f003:**
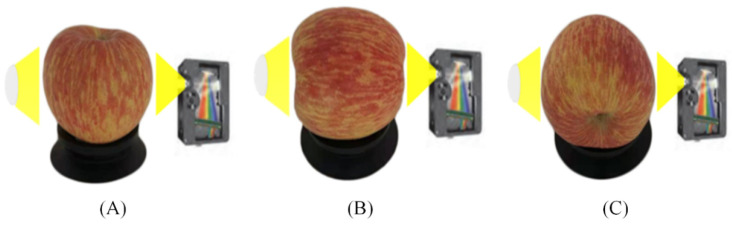
Three different orientations (**A**) O1: Apple stem-calyx axis vertical, stem upward; (**B**) O2: Apple stem-calyx axis horizontal, stem towards light source; (**C**) O3: Apple stem-calyx axis horizontal and parallel to the moving direction of the conveyor belt, calyx in front.

**Figure 4 foods-10-02983-f004:**
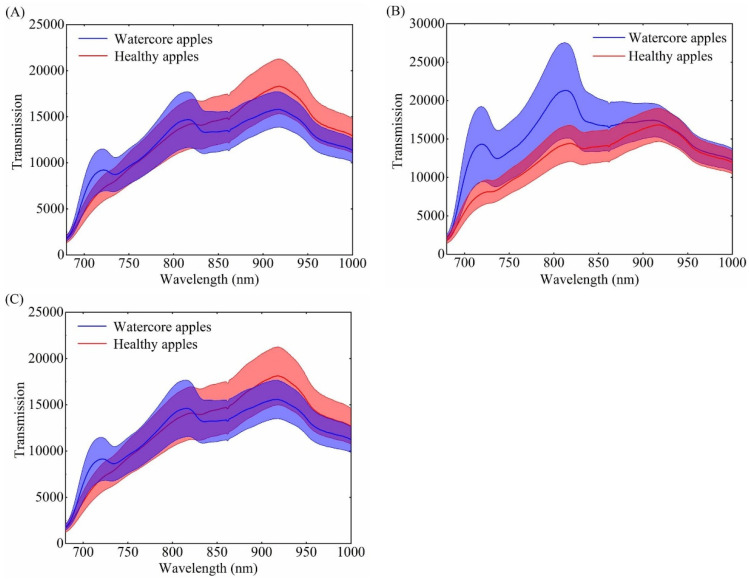
Mean full-transmittance spectra (solid line) and standard deviation (shadow region) of healthy apples and watercore apples. (**A**–**C**) Stand for the original spectra in the O1, O2, and O3 orientations, respectively.

**Figure 5 foods-10-02983-f005:**
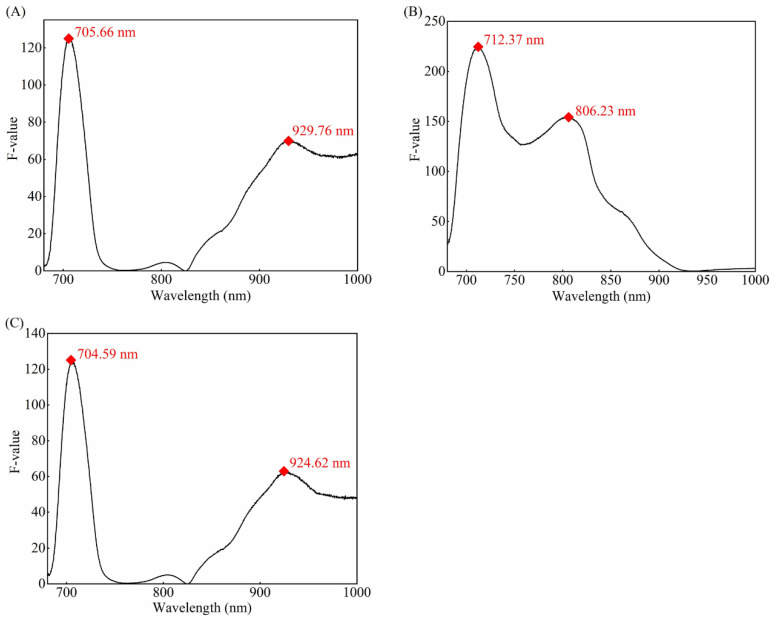
F-values of Vis/NIR full-transmittance spectra between healthy and watercore apples. (**A**) F-values curve of O1 orientation; (**B**) F-values curve of O2 orientation; (**C**) F-values curve of O3 orientation.

**Figure 6 foods-10-02983-f006:**
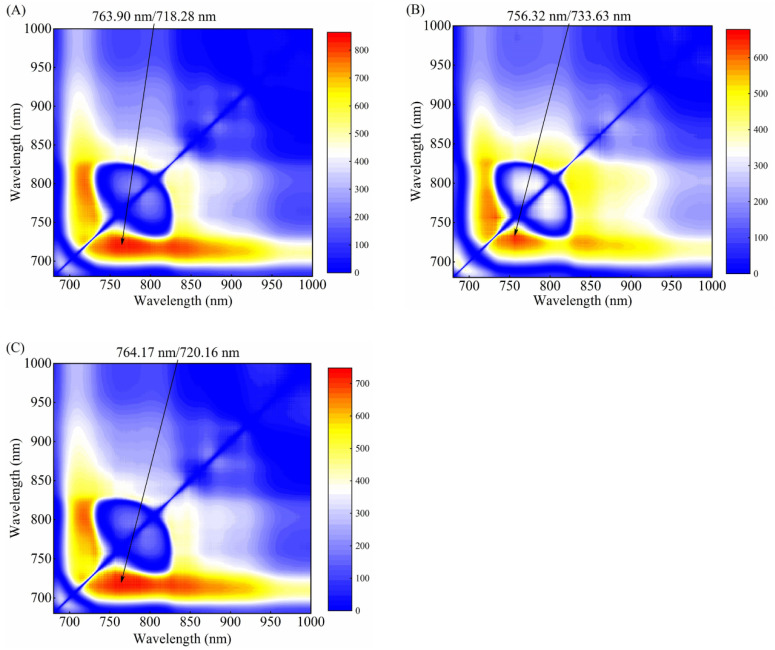
F-values for determining the optimal Vis/NIR full-transmittance spectra band-ratio between healthy and watercore apples. (**A**) O1 orientation spectra; (**B**) O2 orientation spectra; (**C**) O3 orientation spectra.

**Figure 7 foods-10-02983-f007:**
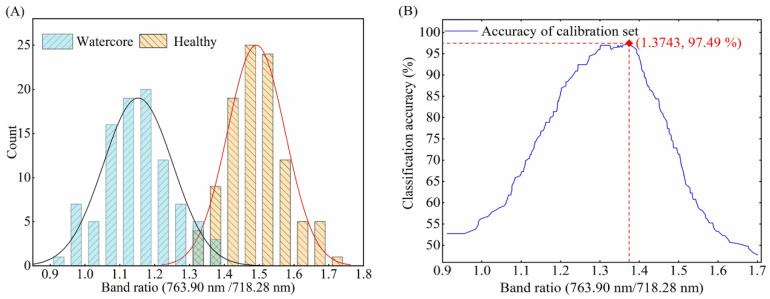
(**A**) Distribution of 763.90 nm/718.28 nm band ratio for healthy and watercore apples spectra in the O1 detection orientation and (**B**) variation of classification accuracy for the calibration set.

**Figure 8 foods-10-02983-f008:**
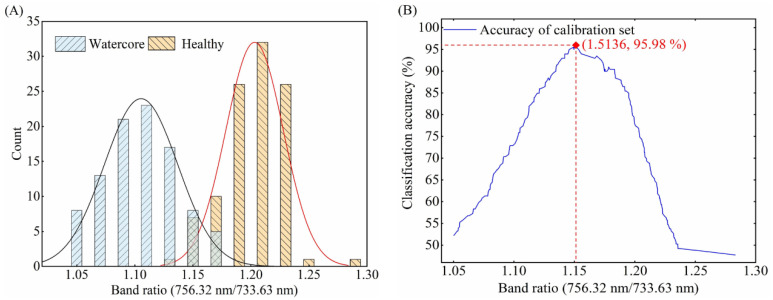
(**A**) Distribution of 756.32 nm/733.63 nm band ratio for healthy and watercore apples spectra in the O2 detection orientation and (**B**) variation of classification accuracy for the calibration set.

**Figure 9 foods-10-02983-f009:**
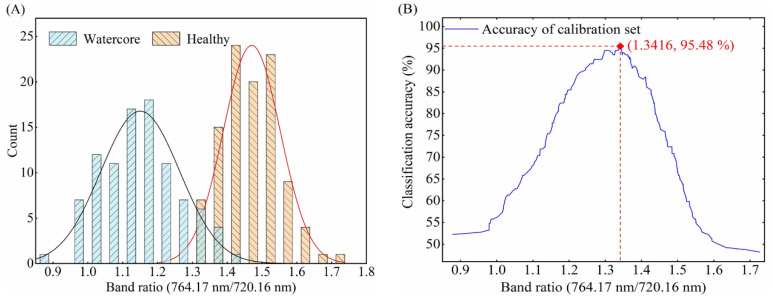
(**A**) Distribution of 764.17 nm/720.16 nm band ratio for healthy and watercore apples spectra in the O3 detection orientation and (**B**) variation of classification accuracy for the calibration set.

**Table 1 foods-10-02983-t001:** Class assignment and partition of sample sets.

Sample Class	No. of Samples	Calibration Set	Prediction Set	Assigned Class
Healthy samples	138	104	34	1
Samples with watercore	127	95	32	−1

**Table 2 foods-10-02983-t002:** The classification results of LS-SVM model established based on full wavelengths.

Detection Orientation	Parameter (γ, σ^2^)	Classification Accuracy of Calibration Set (%)			Classification Accuracy of Prediction Set (%)		
		Total	Healthy	Watercore	Total	Healthy	Watercore
O1	10.8; 4221.8	96.98	96.12	97.89	95.45	100	90.62
O2	4756.5; 6913.9	96.98	97.12	96.84	90.91	100	81.25
O3	75,000.4; 31,321.5	100	100	100	98.48	100	96.87

**Table 3 foods-10-02983-t003:** The classification results of LS-SVM model established based on characteristic wavelengths.

Detection Orientation	Characteristic Wavelength (nm)	Parameter (γ, σ^2^)	Classification Accuracy of Calibration Set (%)			Classification Accuracy of Prediction Set (%)		
			Total	Healthy	Watercore	Total	Healthy	Watercore
O1	705.66; 929.76	3.9; 0.2	97.99	99.04	96.84	93.94	94.12	93.75
O2	712.37; 806.23	268.3; 0.3	92.96	94.23	91.58	89.39	97.06	81.25
O3	704.59; 924.62	8.2; 0.5	97.99	100	95.79	98.48	100	96.87

**Table 4 foods-10-02983-t004:** Classification results based on the threshold value of the optimal band ratio.

Detection Orientation	Threshold Value	Classification Accuracy of Calibration Set (%)			Classification Accuracy of Prediction Set (%)		
		Total	Healthy	Watercore	Total	Healthy	Watercore
O1	1.37637	97.49	95.19	100	95.45	94.12	96.87
O2	1.15155	95.98	97.12	94.74	89.39	100	78.12
O3	1.33929	95.48	97.12	93.68	98.48	100	96.87

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
