# Peer review of "Online Detection of Watercore Apples by Vis/NIR Full-Transmittance Spectroscopy Coupled with ANOVA Method"

_foods, 2021, doi:10.3390/foods10122983_

Round 1
Reviewer 1 Report
The manuscript "Online detection of watercore apples by Vis/NIR full-transmittance spectroscopy coupled with ANOVA method" offers an interesting approach to food analysis. The manuscript is suitable for publication after minor revision. Comments are listed below:
-Please define the abbreviations used in the abstract.
Line 34- What is the SSC content?
Line 84- bacterial names should be written in italics.
Line 151- why is the number of spectra collected different for each sample?
Line 158 - What software was used for ANOVA analysis? Were the average spectra included in the ANOVA analysis?
Line 196- How many bans were used for the analysis? In line 155 you defined 1185 bands, while in line 196 you talk about 1145 bands.
Line 208- How were the apples identified as healthy or water core samples?
-Line 217- "Near-infrared radiation has the absorption, reflection and transmission interaction with the tested sample, and the change of energy can reflect information of the detected substance[54]." What about the UV/VIS range? The following paragraph also mentions wavelengths from the UV/VIS range?
-Line 223- "The transmittance spectral intensity of healthy samples increased from 680 nm and reaches the peak at about 920 nm." How can you explain this?
-Section 3.2. Raw spectra features includes no comparison with the available literature.
Line 276-"However, these selected bands cannot be explained by the chemical bonds within the molecular structures of the two groups of samples."-How would the present result be explained?
Author Response
The manuscript "Online detection of watercore apples by Vis/NIR full-transmittance spectroscopy coupled with ANOVA method" offers an interesting approach to food analysis. The manuscript is suitable for publication after minor revision. Comments are listed below:
Response:
We appreciate your positive comments on this research and thank for constructive suggestions to improve the manuscript.
Please define the abbreviations used in the abstract.
Response: We have defined all abbreviations used in the abstract.
Line 34- What is the SSC content?
Response: SSC is the abbreviation of soluble solids content. In fruit quality analysis, SSC mainly refers to sugar content. Full name of SSC has been added in the revised version (line 36)
Line 84- bacterial names should be written in italics.
Response: It has been revised in the revised version (line 85).
Line 151- why is the number of spectra collected different for each sample?
Response: It is because of size of each sample is different. The corresponding explanation has been added in the revised version (line 151)
Line 158 - What software was used for ANOVA analysis? Were the average spectra included in the ANOVA analysis?
Response: ANOVA analysis was performed by MATLAB 2018b (The Math Works, Natick, MA, USA). In this study, the average spectrum of each sample was used as input for ANOVA analysis. The corresponding explanation has been added in the revised version (line 159-line 161)
Line 196- How many bands were used for the analysis? In line 155 you defined 1185 bands, while in line 196 you talk about 1145 bands.
Response: A total of 1185 bands were used for analysis. This error has been corrected in the revised version (line 192)
Line 208- How were the apples identified as healthy or watercore samples?
Response: Some explanations have been added in the revised version (line 109-line 112)
Line 217- "Near-infrared radiation has the absorption, reflection and transmission interaction with the tested sample, and the change of energy can reflect information of the detected substance[54]." What about the UV/VIS range? The following paragraph also mentions wavelengths from the UV/VIS range?
Response: In this study, only Vis/NIR spectral data with the range of 680~1000 nm is used. It has been mentioned in this article.
Line 223- "The transmittance spectral intensity of healthy samples increased from 680 nm and reaches the peak at about 920 nm." How can you explain this?
Response: The corresponding explanation has been added in the revised version (line 219-line 223).
Section 3.2. Raw spectra features includes no comparison with the available literature.
Responses: The corresponding explanation has been added in the revised version (line 219-line 223).
Line 276-"However, these selected bands cannot be explained by the chemical bonds within the molecular structures of the two groups of samples."-How would the present result be explained?
Responses:ANOVA is a statistical method, which is used to analyze the differences between groups. F-values of a one-way ANOVA were used to find the wavelengths representing statistically significant differences for two groups. The selected bands were determined by the highest F-value, indicating that two groups of samples have significant differences in the selected bands. Some statements have been revised in the revised version (line 274-line 277).

Reviewer 2 Report
Dear authors,
I have found the present study to be very innovative and interesting, and the approach you used might be of practical significance for similar purposes, for other types of foodstuffs. The only thing that I would really like added to the papers is explanation why the variables that you found as influential, are not related nor can provide information about the molecular structure of the samples. I absolutely could not understand this point, and I would like this explanation, not just citations and reference to another work. Apart from this, I would also need to recommend editing of English language.
Best regards
Author Response
Dear authors,
I have found the present study to be very innovative and interesting, and the approach you used might be of practical significance for similar purposes, for other types of foodstuffs.
Response:
We appreciate your positive comments on this research and thank for constructive suggestions to improve the manuscript.
- The only thing that I would really like added to the papers is explanation why the variables that you found as influential, are not related nor can provide information about the molecular structure of the samples. I absolutely could not understand this point, and I would like this explanation, not just citations and reference to another work.
Response: The variables were determined by ANOVA (analysis of variance) method. ANOVA is a statistical method, which is used to analyze the differences between groups. F-values of a one-way ANOVA were used to find the wavelengths representing statistically significant differences for two groups. The selected variables were determined by the highest F-value, indicating that two groups of samples have significant differences in the selected variables. There is no doubt that the intensity and trend of spectrum are closely related to the physical and chemical properties of samples. Some statements have been revised in the revised version.
- Apart from this, I would also need to recommend editing of English language.
We revised the whole manuscript carefully to avoid language errors. In addition, we asked several colleagues who are skilled authors to check the English. We believe that the language is now acceptable for the review process.
